# Health care providers' awareness of breastfeeding practice recommendations during COVID-19 pandemic and associated factors in Northwest Ethiopia, 2021: A multicenter study

**Azmeraw Ambachew Kebede[1], Birhan Tsegaw Taye[2], Kindu Yinges Wondie[1]\*, Agumas Eskezia Tiguh[1], Getachew Azeze Eriku[3], Muhabaw Shumye Mihret[1]**

1 Department of Clinical Midwifery, School of Midwifery, College of Medicine and Health Sciences, University of Gondar, Gondar, Ethiopia, 2 Department of Clinical Midwifery, College of Medicine and Health Sciences, Debre Berhan University, Debre Berhan, Ethiopia, 3 Department of Physiotherapy, College of Medicine and Health Sciences, University of Gondar, Gondar, Ethiopia

\* kinduyinges2010@gmail.com

## Abstract

### Background

Prevention of coronavirus disease 2019 (COVID-19) transmission to newborns is one of the basic components of perinatal care in the era of the COVID-19 pandemic. As such, scientific evidence is compulsory for evidence-based practices. However, there was a scarcity of evidence on health care providers' awareness of breastfeeding practice recommendations during the COVID-19 pandemic in Ethiopia, particularly in the study setting.

### Objective

The study aimed at assessing healthcare providers' awareness of breastfeeding practice recommendations during the COVID-19 pandemic and associated factors among healthcare providers in northwest Ethiopia, 2021.

### Methods

A multicenter cross-sectional study was conducted among 405 healthcare providers working in hospitals of Gondar province from November 15, 2020, to March 10, 2021. A simple random sampling technique was employed to select the study subjects. Data were collected via a structured-self-administered questionnaire. EPI INFO version 7.1.2 and SPSS version 25 were used for data entry and analysis respectively. Binary logistic regression analyses were done to identify associated factors and the adjusted odds ratio (AOR) with its 95% confidence interval (CI) at a p-value of <0.05 was used to declare significant association.

### Results

The healthcare providers' awareness of breastfeeding practice recommendations during the COVID-19 pandemic was 40.7% (95% CI: 35.9, 45.6). Working in a tertiary hospital

**Data Availability Statement:** All relevant data are within the manuscript and its Supporting Information files.

**Funding:** The author(s) received no specific funding for this work.

**Competing interests:** The authors have declared that no competing interests exist.

**Abbreviations:** AOR, Adjusted Odds Ratio; CDC, Center for Disease Control and Prevention; CI, Confidence Interval; COR, Crude Odds Ratio; COVID-19, Corona Virus Disease 2019; HCPs, Health Care Providers; SARS COV-2, Severe Acute Respiratory Syndrome Corona Virus 2; SPSS, Statistical Package for Social Science; WHO, World Health Organization.

(AOR = 3.69; 95% CI: 2.24, 6.08), using COVID-19 guideline updates (AOR = 3.34; 95% CI: 2.1, 5.3), being trained on COVID-19 (AOR = 2.78; 95% CI: 1.74, 4.47), owning a smartphone and/or a computer (AOR = 2.26; 95% CI: 1.39, 3.68), and perceiving that COVID-19 is dangerous (AOR = 1.78; 95% CI: 1.05, 3.01) were factors positively associated with healthcare providers' awareness of breastfeeding practice recommendations during the pandemic of COVID-19.

## Conclusion

Only two in five healthcare providers were aware of recommendations on breastfeeding practice during the COVID-19 pandemic and related to information of accessibility information on COVID-19. Therefore, expanding COVID-19 related information through the provision of COVID-19 training and guidelines to all levels of hospitals would improve health care providers' awareness of breastfeeding practice recommendations amid the COVID-19 pandemic.

## Introduction

The postnatal period is a critical time for the prevention of mother-to-child transmission of a coronavirus disease 2019 (COVID-19) [1]. Despite the risk of COVID-19 transmission to the neonate, the benefits of breastfeeding and mother-baby interaction remains essential practice to prevent infection, and promote health and development [2, 3]. Initiation of exclusive breastfeeding within the first hour of birth and skin-to-skin contact with the proper application of preventive measures are integral elements of breastfeeding care during the COVID-19 pandemic [4–6]. Newborns will acquire SARS-CoV-2 (severe acute respiratory system corona virus-2) from their infected mothers, caregivers, and/or the environment [7], mainly through respiratory droplets [8]. The World Health Organization (WHO) promotes breast milk as the best source of infant nutrition regardless of maternal COVID-19 status [4, 9, 10]. Studies have shown that breast milk from mothers infected with COVID-19 does not cause infection and there is no evidence that the virus is transmitted through breastfeeding [11, 12]. In fact, empirical evidence suggests that breastfeeding might protect the baby against COVID-19 infection even babies born to COVID-19 positive women [13–15], although the possibility of vertical transmission of SARS-CoV-2 is inconclusive [16]. Hence, mothers with suspected or confirmed COVID-19 status are strongly encouraged to initiate and continue breastfeeding and should be counseled that the benefit of breastfeeding outweighs the potential risks of transmission as long as there is no evidence of transmission of COVID-19 through breastfeeding [17, 18]. Similarly, the Centers for Disease Control and Prevention (CDC) [8], the United Nations Children's Fund (UNICEF) [6], Save the Children [5], and the American Academy of Pediatrics (AAP) [19, 20] suggest breastfeeding for women with suspected or confirmed COVID-19 status.

The Chinese neonatal novel Coronavirus 2 Expert Group has proposed temporary isolation of newborns at risk for COVID19 [21]. But the latest WHO guidelines indicate that isolation of newborns and their mothers with actual COVID-19 or suspicious COVID19 is unnecessary [4]. Evidence affirmed that isolating newborns for the sake of COVID-19 prevention is harmful [9, 22–25]. As far as the mother is able to care for her baby [4, 26] and the newborn's clinical condition warrants no admission [4, 8], the baby should stay with the mother practicing

skin-to-skin contact, and rooming-in. If the mother is seriously ill and has difficulty directly breastfeeding her baby due to COVID-19 expressed breast milk can be given [25]. Other options like re-lactation, wet nursing, and breastfeeding from donors are safe [9, 25–27]. All women with confirmed or suspected COVID-19, and/or symptomatic women who choose to breastfeed should take appropriate precautions for hand, breast, and respiratory hygiene [4, 21]. Besides, the number of care providers with babies should be minimized [21, 28] and women whose newborns have suspected or confirmed COVID-19 should get breastfeeding counseling, basic psychosocial support, and practical feeding support [27].

Obviously, the COVID-19 pandemic is disrupting the usual life in all sectors all over the world. One of the highly affected COVID-19 affected sectors is maternity services, where there have been significant disruptions and changes in light of service provision. One of the key components of maternity service which require clear guideline for consistent practices across settings breastfeeding practice amid COVID-19 and these guidelines have been indeed prepared based on the best available evidence generated so far. So, generating representative and ongoing evidence remains a mainstay of a roadmap in this perspective [29, 30]. Empirical evidence showcases that there is non-evidence-based and dilemmatic breastfeeding practice in the era of COVID-19 pandemic in resource-limited countries [2]. In addition, healthcare providers may find themselves in ethically difficult situations as they may be going against recommended practices [3]. In this case, perinatal care providers must be familiar with the recommended guidelines and current evidence of global organizations to be able to provide practical, mother-sensitive, and person-centered support [31].

However, empirical evidence on HCP's awareness of breastfeeding care during the pandemic of COVID-19 is scarce. A descriptive cross-sectional survey in India revealed that 54% of the participants were familiar with WHO recommendations for breastfeeding practices. On the other hand, 15% of participants were completely unaware of WHO's recommendations for breastfeeding during the COVID-19 pandemic [32]. Another descriptive cross-sectional survey showed that 80% of nursing officers and 58% of medical officers knew that mothers with suspected or positive COVID-19 can give expressed breast milk for infants [33].

Although Ethiopian practice guidelines for childbirth care during the COVID-19 pandemic are in agreement with the WHO recommendations, there is little evidence regarding awareness of HCPs to breastfeeding practice recommendations during the pandemic of COVID-19 [34]. Thus, investigating HCPs' awareness of WHO recommendations on breastfeeding practices is helpful to identify knowledge gaps and to take corrective actions for consistent implementation of the recommendations and prevention of COVID-19 transmission to neonates. Despite this fact, there was no literature in this regard in Ethiopia to date to the authors' best knowledge. Having this evidence in mind, we conceive and conducted the current scientific study. Therefore, this study aimed at assessing the HCPs' awareness of WHO's recommendations on breastfeeding practice during the pandemic of COVID-19 and associated factors in hospitals of Gondar province, northwest Ethiopia.

## Methods and materials

### Study design, period, and settings

A multicenter cross-sectional study was conducted in Gondar province hospitals, Northwest Ethiopia from November 15, 2020, to March 10, 2021. The Gondar province includes South Gondar, Central Gondar, West Gondar, and North Gondar administrative zones with an estimated 5,137,443 total population. There are 22 Governmental hospitals in the Gondar province (two comprehensive specialized hospitals, one general hospital, and 19 primary hospitals). These hospitals are serving a population of more than 10 million in the province and the

surrounding administrative zones such as the North Wollo, Waghimra, and Wolkait-Tegede, and Setit-Humera.

## Study population and inclusion criteria

All healthcare providers (i.e. midwives, medical doctors, and integrated emergency surgical officers) working at the maternity centers of the selected hospitals were the study population. Healthcare providers who were available during the data collection period and willing to participate were included.

## Sample size determination and sampling procedure

The sample size was determined by using a single population proportion formula considering the following assumptions: 95% confidence level, 50% provider's awareness regarding breastfeeding practice recommendations during the COVID-19 pandemic (since there were no similar studies), and 5% margin of error.

$$n = \frac{(Z\alpha/2)^2 * p(1-p)}{d^2} = n = \frac{(1.96)^2 * 0.5(1-0.5)}{(0.05)2} = 384.$$ By considering a 10% non-response rate, a final sample size of 422 was obtained. Data were collected from 15 hospitals (two comprehensive specialized hospitals, one general hospital, and 12 primary hospitals). The sampling frame was designed after receiving the list of healthcare providers of each hospital. The total sample size was then proportionally allocated to each selected hospital based on the number of perinatal healthcare providers they have. Finally, a simple random sampling technique was used to choose the study subjects.

## Variables of the study

**Dependent variable.** Healthcare providers' awareness of breastfeeding practice recommendations during the COVID-19 pandemic.

**Independent variables.** Socio-demographic variables: age, sex, marital status, religion, educational level, possession of smartphones and/ or computers, watching television (TV), reading newspapers, and monthly income.

Workplace and profession-related variables: working experience, professional category, the type of the hospital, attending COVID-19 training, receiving essential newborn care training, and location of the hospital.

**Measurements.** The awareness of HCPs to breastfeeding practice recommendations during the COVID-19 pandemic was measured as a composite of 10 questions having a "Yes" or "No" response. A score of "1" was given for "Yes" and a score of "0" was given for "No". Consequently, a maximum of 10 and a minimum of 0 scores were obtained. After that, the mean value of the summative score was computed and determined (5.76). Using the mean awareness score as a cut-off point, HCPs who scored above the mean value were considered as being "aware" and those who scored just or below the mean score were considered as being "not aware". Likewise, the HCPs' awareness of breastfeeding practice recommendations during the COVID-19 pandemic was dichotomized as being aware (which was coded as "1") and not aware (which was coded as "0") [33].

## Data collection tools, methods, and procedures

A structured questionnaire was prepared based on the WHO breastfeeding practice recommendations and related literature [4, 9, 10, 24, 25, 27]. A group of researchers has seen the suitability of the questionnaire and the alpha coefficient was used to measure the reliability of the tool, which became 0.803. Data were collected using a self-administered questionnaire. The

questionnaire comprises socio-demographic characteristics, professional and work-related characteristics, and questions assessing the healthcare providers' awareness of breastfeeding practice recommendations during the COVID-19 pandemic. A total of 20 individuals have participated in the data collection and supervision process. That is 15 diploma and 5 BSc midwives.

### Data quality control

Before the actual data collection, a pretest was conducted on 5% of the calculated sample size outside of the study area and data from the pretest was not included in the main study. A three days' training was given on the overall data collection process and safety measures during the data collection. The questionnaire was checked for completeness by the principal investigator and the supervisors throughout the data collection process.

### Data processing and analysis

Data were entered using EPI INFO version 7.1.2 and analyzed using SPSS version 25. Frequency tables, texts, and graphs were used to present descriptive and analytic statistics. Numbers and percentages were used to describe categorical variables. The binary logistic regression (bivariable and multivariable) model was fitted. Variables with a p-value of < 0.25 in the bivariable regression were entered into the multivariable logistic regression analysis and the adjusted odds ratio (AOR) with its 95% CI and a p-value of ≤ 0.05 was used to declare a significant association with awareness of breastfeeding practice recommendations during the COVID-19 pandemic. Multicollinearity assumption was checked and it was acceptable with a variance inflation factor of <10.

### Ethical considerations

Ethical clearance was obtained from the Institutional Ethical Review Board (IRB) of the University of Gondar (**Reference number**: **V/P/RCS/05/413/2020**) and a letter of permission for administrative approval was acquired from each selected hospital. Written informed consent was taken from each of the study participants after a clear explanation of the aim of the study was given.

## Results

### Socio-demographic characteristics

A total of 405 healthcare providers were included, with a 95.97% response rate. About two hundred seventy-one (66.9%) of the study participants were males and a greater number (59.5%) of HCPs were married. The mean age of the study participants was 28.4 years (SD ±4.7). The mean average monthly income of HCPs was 5862.6 (SD ±1948.5) Ethiopian Birr (ETB) (**Table 1**).

### Workplace and profession-related characteristics

Of the total HCPs, two hundred forty-three (60%) were BSc midwifery professionals and 54.8% of them were working in urban hospitals. About two-thirds (65.4%) of them had more than three years of working experience (**Table 2**).

### Awareness of breastfeeding practice recommendations during COVID-19

The healthcare providers' awareness of breastfeeding practice recommendations during the COVID-19 pandemic was 40.7% (95% CI: 35.9, 45.6). When HCPs were asked about the

**Table 1. Socio-demographic characteristics of healthcare providers in hospitals of Gondar province, 2021 (n = 405).**

| Characteristics | Frequency | Percentage (%) |
|---|---|---|
| Age in years | | |
| ≤ 25 | 85 | 21.0 |
| 26–30 | 246 | 60.7 |
| ≥ 31 | 74 | 18.3 |
| Sex | | |
| Male | 271 | 66.9 |
| Female | 134 | 33.1 |
| Current marital status | | |
| Single | 164 | 40.5 |
| Married | 241 | 59.5 |
| Religion | | |
| Orthodox Christian | 380 | 93.8 |
| Other☞ | 25 | 6.2 |
| Having smart phone or computer | | |
| Yes | 256 | 63.2 |
| No | 149 | 36.8 |
| Average monthly income in Ethiopian birr | | |
| ≤ 5, 000 | 140 | 34.6 |
| 5, 001–10, 000 | 238 | 58.7 |
| ≥ 10001 | 27 | 6.7 |

Note: ☞Muslim, catholic Christian & protestant Christian.

breastfeeding practice recommendations during the COVID-19 pandemic, more than half (51.4%) knew that skin-to-skin contact is encouraged, but two-thirds (66.4%) of them did not know rooming-in is recommended. About 222 (54.8%) of HCPs knew that the benefit of breastfeeding outweighs its risk (**Table 3**).

## Factors associated with healthcare providers' awareness of breastfeeding practice recommendations during the COVID-19 pandemic

Multivariable logistic regression analysis showed that healthcare providers who were working in a tertiary hospital, had a smartphone and/or computer, perceiving that COVID-19 is dangerous, trained on COVID-19, and following the WHO and/or CDC COVID-19 updates were found to be significantly associated with HCPs' awareness of breastfeeding practice recommendations during the COVID-19 pandemic.

HCPs working in a tertiary hospital were 3.69 (AOR = 3.69; 95% CI: 2.24, 6.08) times more likely to be aware of breastfeeding practice recommendations during the COVID-19 pandemic than those HCPs working in a primary hospital. Besides, HCPs with a smartphone and/or computer were 2.26 times more likely to be aware of breastfeeding practice recommendations during the COVID-19 pandemic than HCPs without a smartphone and/or computer (AOR = 2.26; 95% CI: 1.39, 3.68).

Similarly, the odds of being aware of breastfeeding practice recommendations during the COVID-19 pandemic was 1.78 (AOR = 1.78; 95% CI: 1.05, 3.01) times higher among HCPs who perceived that COVID-19 is dangerous compared with their counterparts. Moreover, HCPs who received COVID-19 training had 2.78 times more odds of awareness of

**Table 2. Working place, profession, and COVID-19 related characteristics of study participants in hospitals of Gondar province, 2021 (n = 405).**

| Characteristics | Frequency | Percentage (%) |
|---|---|---|
| Professional category | | |
| Diploma midwifery | 118 | 29.1 |
| Degree midwifery | 243 | 60 |
| Master's midwifery | 25 | 6.2 |
| Others☞ | 19 | 4.7 |
| Hospital category | | |
| Primary hospital | 176 | 43.5 |
| General hospital | 68 | 16.8 |
| Comprehensive specialized hospital | 161 | 39.8 |
| Hospital location | | |
| Urban | 222 | 54.8 |
| Semi-urban | 183 | 45.2 |
| Work experience in years | | |
| ≤ 2 | 140 | 34.6 |
| 3–5 | 209 | 51.6 |
| > 5 | 56 | 13.8 |
| COVID-19 is dangerous | | |
| Yes | 299 | 73.8 |
| No | 106 | 26.2 |
| Human milk contains viable SARS-CoV-2 | | |
| Yes | 292 | 27.9 |
| No | 113 | 72.1 |
| Attending COVID-19 training | | |
| Yes | 234 | 57.8 |
| No | 171 | 42.2 |
| Following the WHO and or/CDC COVID-19 updates | | |
| Yes | 186 | 45.9 |
| No | 219 | 54.1 |
| Training on essential newborn care | | |
| Yes | 156 | 38.5 |
| No | 249 | 61.5 |
| Using the internet as a source of information about COVID-19 | | |
| Yes | 232 | 57.3 |
| No | 173 | 42.7 |
| COVID-19 preventive measures are effective | | |
| Yes | 109 | 26.9 |
| No | 296 | 73.1 |

Note: ☞ General practitioner, Obstetrics and Gynecology Resident, Integrated Emergency Surgery Officer, Obstetrician.

breastfeeding practice recommendations than the untrained counterparts (AOR = 2.78; 95% CI: 1.74, 4.47). Furthermore, the likelihood of awareness of breastfeeding practice recommendations during the COVID-19 pandemic was three times higher among HCPs who were regularly followed the WHO and CDC COVID-19 updates compared with that of those who did not so (AOR = 3.34; 95% CI: 2.10, 5.30) (**Table 4**).

**Table 3. The WHO breastfeeding practice recommendations during COVID-19 pandemic in suspected or confirmed COVID-19 cases, Gondar province (n = 405).**

| Characteristics | Frequency | Percentage (%) |
|---|---|---|
| Early skin to skin contact is recommended | | |
| Yes | 208 | 51.4 |
| No | 197 | 48.6 |
| The mother should be told that the benefit of breastfeeding outweighs the potential risks of COVID-19 transmission | | |
| Yes | 222 | 54.8 |
| No | 183 | 45.2 |
| The umbilical cord should be clamped as usual | | |
| Yes | 312 | 77 |
| No | 93 | 23 |
| Breastfeeding should be initiated within one hour after delivery | | |
| Yes | 296 | 73.1 |
| No | 109 | 26.9 |
| Direct breastfeeding is recommended | | |
| Yes | 214 | 52.8 |
| No | 191 | 47.2 |
| A mother with COVID-19 can give expressed breast milk if she is too unwell to directly breastfeed | | |
| Yes | 299 | 73.8 |
| No | 106 | 26.2 |
| A COVID-19 suspected or confirmed mother should take routine precaution measures | | |
| Yes | 85 | 21 |
| No | 320 | 79 |
| Donor human milk or wet nursing can be used if a woman is severely infected with COVID-19 | | |
| Yes | 149 | 36.8 |
| No | 256 | 63.2 |
| Rooming-in is recommended? | | |
| Yes | 136 | 33.6 |
| No | 269 | 66.4 |
| Psychological or practical breastfeeding support should be given to the mother if she prefers to breastfeed | | |
| Yes | 241 | 59.5 |
| No | 164 | 40.5 |

## Discussion

Proper counseling to prevent COVID-19 in newborns and the implementation of precautions by health care providers are crucial components of perinatal care during the COVID-19 pandemic [7, 14]. In this context, the assessment of HCPs' awareness of breastfeeding practice recommendations during the COVID-19 pandemic plays a key role. Thus, this cross-sectional study was conducted to assess the awareness of breastfeeding practice recommendations during the COVID-19 pandemic and associated factors among HCPs in Gondar province, Northwest Ethiopia. Accordingly, two-fifths of the HCPs were aware of the breastfeeding practice recommendations during the COVID-19 pandemic. Working in a tertiary hospital, owning a smartphone and/or a computer, perceiving that COVID-19 is dangerous, being trained on

**Table 4. Logistic regression analysis of factors associated with healthcare providers' awareness of the breastfeeding practice recommendations during the COVID-19 pandemic in Gondar province, 2021 (n = 405).**

| Variables | Awareness to breastfeeding practice recommendation | | COR (95% CI) | AOR (95% CI) |
|---|---|---|---|---|
| | Aware | Not aware | | |
| Professional category | | | | |
| Diploma midwifery | 39 | 79 | 1 | |
| Degree midwifery | 106 | 137 | 1.57(0.99, 2.48) | 0.91(0.52, 1.62) |
| Master's midwifery | 15 | 10 | 3.04(1.25, 7.38) | 1.13(0.37, 3.44) |
| Others☞ | 5 | 14 | 0.72(0.24, 2.75) | 0.63(0.19, 2.62) |
| Hospital category | | | | |
| Primary hospital | 48 | 128 | 1 | |
| General hospital | 23 | 45 | 1.36 (0.75, 2.49) | 1.33(0.69, 2.54) |
| Comprehensive specialized hospital | 94 | 67 | 3.74 (2.37, 5.91) | 3.69(2.24, 6.08)*** |
| Hospital location | | | | |
| Urban | 115 | 107 | 2.86 (1.88, 4.34) | 0.76 (0. 26, 2.22) |
| Semi-urban | 50 | 133 | 1 | |
| COVID-19 is dangerous | | | | |
| Yes | 131 | 168 | 1.65 (1.04, 2.64) | 1.78(1.05, 3.01)* |
| No | 34 | 72 | 1 | |
| Having smart phone and/or computer | | | | |
| Yes | 119 | 137 | 1.94 (1.27, 2.98) | 2.26(1.39, 3.68)** |
| No | 46 | 103 | 1 | |
| Using internet as a source of information about COVID-19 | | | | |
| Yes | 109 | 123 | 1.85 (1.23, 2.79) | 1.25 (0.7, 2.21) |
| No | 56 | 117 | 1 | |
| Attending COVID-19 training | | | | |
| Yes | 117 | 117 | 2.56 (1.68, 3.9) | 2.78(1.74, 4.47)*** |
| No | 48 | 123 | 1 | |
| Following the WHO and or/CDC COVID-19 updates | | | | |
| Yes | 99 | 87 | 2.64(1.76, 3.97) | 3.34(2.1, 5.3)*** |
| No | 66 | 153 | 1 | |

Notes

* P ≤ 0.05

**P ≤ 0.01

*** P ≤ 0.001; ☞ General practitioner, Obstetrics and Gynecology Resident, and Integrated Emergency Surgical Officers; CDC-Centre for Disease Control and Prevention; WHO—World Health Organization.

COVID-19, and using the WHO and/or CDC COVID-19 updates were significant predictors of HCPs' awareness of breastfeeding practice recommendations during the COVID-19 pandemic.

This study found that the HCPs' awareness of the breastfeeding practice recommendations during the COVID-19 pandemic was 40.7%, which is lower than a report from India-54.1% [32]. The discrepancy might be due to differences in the study population. The Indian study was done among specialists such as pediatricians (85.1%), and obstetricians (14.9%). In contrast, the majority of our study participants were midwifery professionals and BSc degree holders. Specialists are top consultants in inpatient care, so better knowledge from specialists is expected. In this regard, it is highly recommended that all healthcare providers, regardless of education level, be aware of important breastfeeding recommendations during the COVID-19 pandemic. In this study, a higher awareness of breastfeeding practice recommendations during the COVID-19 was identified in some of the questions by HCPs. Thus, nearly three-quarters (73.8%) of HCPs were aware of a mother with COVID-19 can give expressed breast milk if she is too ill to directly breastfeed her newborn. Because it is highly recommended that a woman can give her baby expressed breast milk even if she is too ill and unable to breastfeed directly [25]. Besides, about 73.1% of HCPs aware of breastfeeding should be initiated within one hour after delivery. As long as a woman can breastfeed her newborn, breastfeeding should be initiated within 30 to 60 minutes as usual regardless of the woman's COVID-19 status. On the other hand, only a third (33.6%) and one-fifth (21%) of HCPs recognize rooming-in and the need for routine precaution for women suspected or confirmed COVID-19. This urges concerned bodies and health policymakers to provide updated information and arrange training opportunities for HCPs regarding breastfeeding practices during the COVID-19 pandemic regularly. This will help HCPs to strengthen their existing knowledge and fill the gaps that they have regarding cautions taken during breastfeeding, thereby preventing unnecessary actions taken.

This study revealed that HCPs working in the tertiary hospital were 3.8 times more likely to be aware of breastfeeding practice recommendations during the COVID-19 pandemic compared with HCPs working in the primary hospital. The possible explanation might be that most of the tertiary hospitals in Ethiopia are University hospitals, usually are the center of excellence, where most training and research conferences take place.

Likewise, owning a smartphone and/or a computer was also found to be significantly affecting the HCPs' awareness of breastfeeding practice recommendations during the COVID-19 pandemic. Hence, HCPs who had a smartphone and/or a computer were two times to be aware of breastfeeding practice recommendations during the COVID-19 pandemic as compared to those who did not own a smartphone and/or a computer. The possible reason might be that having a smartphone and/or computer at hand could increase access to social media utilization and organizational websites, which were found to increase HCPs' knowledge so far [35]. The WHO released recommendation updates in the form of guidelines [7, 8], scientific brief [15], position statement [4], questions and answers [23], and news [25] for healthcare providers that can be accessed through the organizations' website using the internet.

Similarly, the odds of being aware of breastfeeding practice recommendations during the COVID-19 pandemic were 1.8 times higher among HCPs who perceived that COVID-19 is dangerous than their counterparts. The possible explanation is that the more perceived risk of a disease one has, the better his enthusiasm would be to know about it. Attending COVID-19 training among HCPs who perceive that COVID-19 is dangerous and those who did not so was 75.2% and 24.8% respectively. Moreover, following the WHO and or CDC COVID-19 updates was 69.9% and 30.1% among those who perceive that COVID-19 is dangerous and those who did not respectively.

Moreover, taking a COVID-19 training was found to increase HCPs' awareness of the breastfeeding practice recommendations. Thus, HCPs who had taken training on COVID-19 were 2.8 times to be aware of breastfeeding practice recommendations during the COVID-19

pandemic than their counterparts. This finding is supported by previous studies [35–40]. The COVID-19 training given by the Ethiopian public health institute was based on the national COVID-19 guideline for the management of pediatric patients during the COVID-19 pandemic which addresses the issue of breastfeeding and newborn care [41].

Furthermore, HCPs who were following the WHO and/or CDC COVID-19 guideline updates were 2.6 times more likely to be aware of breastfeeding practice recommendations during the COVID-19 pandemic as compared to their counterparts. The finding is supported by studies in Ethiopia [35, 42]. The WHO and CDC released guidelines, scientific briefs, and position papers on breastfeeding and care of newborn babies for healthcare providers [4, 9, 18, 26]. So that, HCPs who use these guidelines might be aware of the breastfeeding practice recommendations.

## Limitations of the study

We are pleased to acknowledge some of the limitations of the current study. First, there was no previously standardized and validated tool to assess the HCPs' awareness of breastfeeding practice recommendations during the COVID-19 pandemic. As the area is not well studied, we didn't find adequate studies to compare and contrast our findings with others, which made our discussion shallow. Second, due to the cross-sectional nature of the study design, inferring the cause-effect relationship between the dependent and independent variables might be impossible. Third, it is impossible to ensure the validity of the responses since the questionnaire was self-administered. However, the study participants were informed to fill the data honestly and their true response is crucial for the success of the study. Despite these limitations, our findings provide valuable information about HCPs' awareness of breastfeeding practice recommendations during the COVID-19 pandemic.

## Conclusion

The HCPs' awareness of the breastfeeding practice recommendations during the COVID-19 pandemic was found to be low and it was positively associated with working in a tertiary hospital, following the WHO and/or CDC COVID-19 update, being trained on COVID-19, owning a smartphone, and/or a computer, and perceiving that COVID-19 is dangerous. A more rigorous update communication of practice recommendations on breastfeeding care among COVID-19 suspected and confirmed mother-baby dyads needs to be adopted through training especially towards lower-level HCPs.

## Supporting information

**S1 File. English version of the questionnaire.**
(DOCX)

**S2 File. SPSS dataset.**
(SAV)

## Acknowledgments

We would like to thank the University of Gondar for providing study ethical clearance to conduct this study. Our gratitude also goes to all data collectors and study participants. We are glad to Hospitals in Gondar province for writing permission letter.

## Author Contributions

**Conceptualization:** Azmeraw Ambachew Kebede.

**Data curation:** Azmeraw Ambachew Kebede, Birhan Tsegaw Taye, Kindu Yinges Wondie, Agumas Eskezia Tiguh, Getachew Azeze Eriku, Muhabaw Shumye Mihret.

**Formal analysis:** Azmeraw Ambachew Kebede, Birhan Tsegaw Taye, Kindu Yinges Wondie, Agumas Eskezia Tiguh, Getachew Azeze Eriku, Muhabaw Shumye Mihret.

**Investigation:** Azmeraw Ambachew Kebede, Birhan Tsegaw Taye, Muhabaw Shumye Mihret.

**Methodology:** Azmeraw Ambachew Kebede, Birhan Tsegaw Taye, Kindu Yinges Wondie, Agumas Eskezia Tiguh, Getachew Azeze Eriku, Muhabaw Shumye Mihret.

**Validation:** Azmeraw Ambachew Kebede, Birhan Tsegaw Taye, Kindu Yinges Wondie, Agumas Eskezia Tiguh, Getachew Azeze Eriku, Muhabaw Shumye Mihret.

**Visualization:** Azmeraw Ambachew Kebede, Birhan Tsegaw Taye, Kindu Yinges Wondie, Agumas Eskezia Tiguh, Getachew Azeze Eriku, Muhabaw Shumye Mihret.

**Writing – original draft:** Kindu Yinges Wondie.

**Writing – review & editing:** Azmeraw Ambachew Kebede, Birhan Tsegaw Taye, Kindu Yinges Wondie, Agumas Eskezia Tiguh, Getachew Azeze Eriku, Muhabaw Shumye Mihret.

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
