## [Decision Letter · Decision Letter 0]

10 Nov 2021

PONE-D-21-32452Health care providers’ awareness of breastfeeding practice recommendations during COVID-19 pandemic and associated factors in Northwest Ethiopia, 2021: a multicenter studyPLOS ONE

Dear Dr. Wondie,

Thank you for submitting your manuscript to PLOS ONE. After careful consideration, we feel that it has merit but does not fully meet PLOS ONE’s publication criteria as it currently stands. Therefore, we invite you to submit a revised version of the manuscript that addresses the points raised during the review process.

We look forward to receiving your revised manuscript.

Kind regards,

Rohit Ravi, Ph.D.

Academic Editor

PLOS ONE

Journal Requirements:

Reviewers' comments:

Reviewer's Responses to Questions

**Comments to the Author**

1. Is the manuscript technically sound, and do the data support the conclusions?

Reviewer #1: Yes

Reviewer #2: Partly

Reviewer #3: Yes

2. Has the statistical analysis been performed appropriately and rigorously? 

Reviewer #1: Yes

Reviewer #2: Yes

Reviewer #3: No

3. Have the authors made all data underlying the findings in their manuscript fully available?

Reviewer #1: Yes

Reviewer #2: Yes

Reviewer #3: Yes

4. Is the manuscript presented in an intelligible fashion and written in standard English?

Reviewer #1: Yes

Reviewer #2: Yes

Reviewer #3: No

5. Review Comments to the Author

Reviewer #1: Regarding the data availability, authors agreed its available, is it that SPSS file at the end of document?

Table 1 title is mentioned twice, please check.

How p value of 0.2 is set for multivariable regression?

Reviewer #2: 1) Technical sections such as Materials & Methods section needs to expanded for the reader to completely grasp the concept, For example, Clear definition of what an HCP (Health Care Professional) is within the confines of the study needs to be stated, there is a need for explicitly stating why Gondar Province Hospitals were selected as a study setting and why was it only restricted to Government Hospitals - the same needs to be Substantiated with a Rationale, Also For computation of sample size justification for using 50% as provider awareness needs to be clearly stated

2) As per the guidelines enlisted, the Tables within the article should be included directly after the paragraph in which they are first cited

3) The authors need to explicitly state the cut off's of one considered being aware or unaware that were used for computation and drawing inference to improve the understanding of the reader.

4) The Authors to Cross check the Gender disaggregation stated in the Result section and the Appendix Table

5) Adherence to the sequence: Acknowledgments, References, Supporting information captions (if applicable) is required per the guidelines

6) Revision of language and formatting in a couple of places is required, including cross checking of the numbering of the Tables

7) The additional information under the tables are to be included in the the form of a footnote

8) There is a need to recheck and edit the References to meet the Vancouver format guidelines

9) The limitation section, needs to further highlight whether there are any potential bias, confounding variables that would potentially have an impact on the findings and inferences drawn, if any

Reviewer #3: Overall, this manuscript may be useful to know what factors contribute to health care providers' awareness of breastfeeding practice recommendations. This research contributes to the body of knowledge and database.

Few suggestions include:

1. In abstract section, the data collection duration need to be corrected.

2. Observations that the breast feeding practice recommendations are being discussed for mothers suspected of or diagnosed with Covid-19 came very late in the process. It should be emphasized right from the start, starting with the objectives.

3. Under data processing and analysis, it is stated that those variables with p value 0.2 were included in logistic regression. What is the basis for taking such value? This raises a concern for overall regression analysis. This needs to be relooked.

4. Under the results section, under factors associated- it was mentioned that good or bad awareness. This is a wrong use of terminologies as there is nothing like good or bad. But instead, there could be certain factors which were associated with more likelihood of awareness. The results may be rewritten for better appropriate use of terminology.

5. The manuscript must be extensively revised for language and grammar.

6. We understand that there is paucity of evidence in this area and this limited the discussion in comparison with other studies. However, the discussion currently is giving results more quantitatively rather than a perspective that can be a little qualitative. May be a discussion around what the guidelines say and what is observed in the current study could be talked a little more. Apart from just the predictors, it may also be good to understand awareness around which indicators based on the questions used in the questionnaire is low and needs more attention. This will tell the readers what areas need more attention for creating awareness among health care providers.

6. PLOS authors have the option to publish the peer review history of their article (what does this mean?). If published, this will include your full peer review and any attached files.

Reviewer #1: No

Reviewer #2: No

Reviewer #3: No

---

## [Author Response · Author response to Decision Letter 0]

15 Nov 2021

Have been attached as a separate file.

---

## [Editor Report · Decision Letter 1]

17 Nov 2021

Health care providers’ awareness of breastfeeding practice recommendations during COVID-19 pandemic and associated factors in Northwest Ethiopia, 2021: a multicenter study

PONE-D-21-32452R1

Dear Dr. Wondie,

We’re pleased to inform you that your manuscript has been judged scientifically suitable for publication and will be formally accepted for publication once it meets all outstanding technical requirements.

Kind regards,

Rohit Ravi, Ph.D.

Academic Editor

PLOS ONE

Additional Editor Comments (optional):

Dear Authors, The changes made to the manuscipt are satisfactory.
---

## [Editor Report · Acceptance letter]

19 Nov 2021

PONE-D-21-32452R1 

Health care providers’ awareness of breastfeeding practice recommendations during COVID-19 pandemic and associated factors in Northwest Ethiopia, 2021: a multicenter study 

Dear Dr. Wondie:

I'm pleased to inform you that your manuscript has been deemed suitable for publication in PLOS ONE. Congratulations! Your manuscript is now with our production department. 

Kind regards, 

on behalf of

Dr. Rohit Ravi 

Academic Editor

PLOS ONE